# Pseudouridine Synthase RsuA Captures an Assembly Intermediate That Is Stabilized by Ribosomal Protein S17

**DOI:** 10.3390/biom10060841

**Published:** 2020-05-30

**Authors:** Kumudie Jayalath, Sean Frisbie, Minhchau To, Sanjaya Abeysirigunawardena

**Affiliations:** Department of Chemistry and Biochemistry, Kent State University, Kent, OH 44242, USA; kjayalat@kent.edu (K.J.); sfrisbie@kent.edu (S.F.); mto1@kent.edu (M.T.)

**Keywords:** bacterial ribosome assembly, nucleotide modification enzymes, pseudouridine synthase

## Abstract

The ribosome is a large ribonucleoprotein complex that synthesizes protein in all living organisms. Ribosome biogenesis is a complex process that requires synchronization of various cellular events, including ribosomal RNA (rRNA) transcription, ribosome assembly, and processing and post-transcriptional modification of rRNA. Ribosome biogenesis is fine-tuned with various assembly factors, possibly including nucleotide modification enzymes. Ribosomal small subunit pseudouridine synthase A (RsuA) pseudouridylates U516 of 16S helix 18. Protein RsuA is a multi-domain protein that contains the N-terminal peripheral domain, which is structurally similar to the ribosomal protein S4. Our study shows RsuA preferably binds and pseudouridylates an assembly intermediate that is stabilized by ribosomal protein S17 over the native-like complex. In addition, the N-terminal domain truncated RsuA showed that the presence of the S4-like domain is important for RsuA substrate recognition.

## 1. Introduction

Ribosomes are ribonucleoprotein complexes that synthesize proteins in all living organisms. Accurate and efficient synthesis of new ribosomes is critical for cell survival. During in vitro assembly of the 30S ribosome, ribosomal proteins (r-proteins) bind to 16S rRNA hierarchically that arise from the cooperative addition of r-proteins [1,2,3]. Additions of r-proteins, rRNA folding, rRNA modification, and rRNA processing occur concurrently during the in vivo ribosome biogenesis [4]. Various trans-acting factors are required to maintain the synchronicity of these various processes and increase the efficiency and accuracy of ribosome biogenesis. 16S rRNA is modified sequentially from 5′–3′ direction while the ribosome assembly is in progress. Modification enzymes responsible for pseudouridine (Ψ) and m^7^G at positions 516 and 527 (*Escherichia coli* numbering) in the 5′-domain can bind to early assembly intermediates, hence influence the association of late-binding r-proteins. Thermodynamic and kinetic cooperativities between early-binding modification enzymes and r-proteins could alter the 30S assembly energy landscape and direct assembly flux through productive and fast pathways.

Pseudouridine synthase (PUS) enzymes lyse the C-N glycosidic bond in uridine and isomerize to form a C-C glycosidic bond between C1′ and C5 of sugar and base, respectively. All PUS enzymes are classified into five families based on their sequences. Although these five families of PUS enzymes are dissimilar in terms of their sequence, their catalytic domains are structurally similar. However, the peripheral domains of these enzymes vary for different families of pseudouridine synthases, except for some RluA family PUS enzymes that have peripheral domains like those of RsuA family PUS enzymes. Ribosomal small subunit pseudouridine synthase A (RsuA) modifies the U516 of 16S rRNA to a pseudouridine [5,6] (Figure 1C). The Ψ516 is the only pseudouridine residue found in 16S rRNA [7] (Figure 1A,B). Protein RsuA is a multi-domain protein that contains an N-terminal domain (Met 1-Pro 55), a central domain (Tyr 62-Thr 122, Gly 207–Val 231), and a C-terminal domain (Ser 123-Gly 206) [8]. Catalytically essential and highly conserved, Asp 102 residue is located in the central domain [6]. The RsuA N-terminal domain, also known as the S4-like domain, is an α3β4 domain that is structurally similar to the C-terminal domain of ribosomal protein S4 [8,9,10]. Unlike many other pseudouridine synthase enzymes, RsuA lacks a thumb domain that helps to place uridine into the active site of the modification enzyme [8]. Protein RsuA likely binds to an RNA conformation in which U516 is flipped out and projected towards the RsuA active site. RsuA S4-like domain may assist the protein to recognize a favorable conformation for RsuA activity [11]. However, S4-like domain deletion mutant of ribosomal large subunit pseudouridine synthase C (RluC) showed that the enzyme is still capable pseudouridylation even without the presence of the S4-like domain [12]. The S4-like domain is connected to the RsuA core made up of C-terminal and central domains by an unstructured linker sequence that may allow the S4-like domain to sample multiple conformations with respect to the RsuA core domain [8]. The S4-like domain in *E. coli* RsuA X-ray crystal structure extends away from the catalytic domain and adopts the open conformation, whereas in that of *H. influenza*, it is bent towards the catalytic domain to adopt a more compact conformation [8,13].

In eukaryotes, mutant PUS enzymes can cause rare diseases, such as dyskaryosis congenita. Although most bacterial PUS enzymes are non-essential for bacterial survival, bacterial growth defects are observed in RluD deletion strains. Similarly, TruB deletion strains illustrated a strong selection disadvantage against wild type *E. coli* strains [14]. However, these growth defects could be reversed by replacing the wild type RluD and TruB with non-functional mutants, suggesting far more important roles for those PUS enzymes than their enzymatic activity [15]. For instance, TruB showed RNA chaperone activity. Similarly, many rRNA PUS enzymes may have roles other than pseudouridylation, likely related to ribosome biogenesis. This study is focused on the determination of the preferred substrates for RsuA binding and its activity thereby understanding the impact of RsuA on ribosome biogenesis. Furthermore, the importance of RsuA peripheral domain in the binding of the RsuA is also discussed.

## 2. Materials and Methods

### 2.1. Overexpression and Purification of Proteins

RsuA overexpression and purification were carried out as previously described [5]. RsuA overexpression plasmid, pCA24N, was purchased from ASKA plasmid collection (National BioResource Project: NBRP *E. coli* strain) [16]. Site-directed mutagenesis (Agilent, QuickChange) was carried out to remove an additional DNA sequence adjacent to the stop codon and prepared the RsuA: G127C mutant. The mutant RsuA sequence was confirmed with Sanger sequencing. The expression vector, pCA24N with mutant RsuA sequence, was transformed to the BL21 (DE3) competent *E. coli* cells (NEB) using manufacturer suggested protocol. These cells were grown, and RsuA protein was overexpressed, as previously mentioned [5]. The bacterial cells were resuspended in 20 mL of lysis buffer (20 mM K-Hepes, pH 7.6, 1 M KCl, 1 mM TCEP) containing 200 µL of Halt^TM^ protease inhibitor cocktail -100X (Thermo Fisher) and sonicated to lyse the cells. The cell lysate was centrifuged at 10,000 rpm for 15 min and the supernatant was dialyzed against buffer A (20 mM K-Hepes, pH 7.6, 1 M KCl, 1 mM TCEP). Dialyzed cell lysate was passed through a Ni-NTA column (HisTrap^TM^ HP, GE Healthcare) connected to an AKTA start. Proteins bound specifically to the column were eluted using an imidazole gradient created with buffer B (20 mM K-Hepes, pH 7.6, 1 M KCl, 1M imidazole, 1 mM TCEP). Protein RsuA was eluted at 500 mM imidazole. The purity of the protein was confirmed using SDS-PAGE (Appendix A). Purified RsuA protein was dialyzed overnight (three buffer changes) into the storage buffer (80 mM K-Hepes, pH 7.6, 1 M KCl, 6 mM β-mercaptoethanol). Protein was aliquoted, snap-freeze, and stored at −80 °C until further use. Ribosomal proteins (S4, S17, S16, and S20) were overexpressed, purified, and stored at −80 °C, as described in previous literature (Appendix A) [17].

### 2.2. Fluorescence Labeling of RsuA

Fluorescence labeling of RsuA: G127C mutant was carried out as previously described [18]. Purified RsuA was incubated at 20 °C for 30 min before the addition of 6× excess of maleimide-linked Cy5 (GE Healthcare). The reaction mixture was incubated at 20 °C for another 2 h and quenched by adding β-mercaptoethanol (6 mM). The final KCl concentration of the labeling reaction was adjusted to 20 mM before passing through a BioRad Uno S6 column to remove the unreacted dye. Fluorescently labeled RsuA: G127C was eluted with a KCl gradient. Isolated Cy5-labeled RsuA protein was run on an SDS-PAGE gel (Appendix A). The gel was scanned using Typhoon3000 to confirm the Cy5 labeling. The labeled RsuA: G127C was dialyzed overnight into a final storage buffer (80 mM K-Hepes, pH 7.6, 1 M KCl, 6 mM β-mercaptoethanol). The concentration of labeled protein was determined using a NanoDrop (ε650 = 250,000) and stored at −80 °C in a light-tight box.

### 2.3. 16S 5′-Domain RNA Preparation

All *E. coli* 16S 5′-domain rRNAs (*E. coli* numbering, 21-556 nucleotides) used in FRET (Förster resonance energy transfer)- assays contained AAGGACGACACACUUUGGACAGGACACACAGGACACAGG (E5 extension) added to its 3′-end next to helix 3 as previously mentioned [18]. Site-directed mutagenesis using a QuikChange mutagenesis kit (Agilent) was performed to generate 16S helix 18 pseudoknot mutants (G505C, C507G, G505C:G506C) used in this study. Manufacturer suggested protocols were followed for this mutagenesis. The presence of mutations was confirmed by Sanger sequencing (GenScript). These various 5′-domain RNAs were transcribed in vitro and purified using 4% polyacrylamide gel electrophoresis, as described previously. RNAs from gel bands were eluted using the freeze-thaw method, followed by phenol-chloroform extraction and ethanol precipitation. RNAs were dissolved in TE buffer (10 mM Tris-HCl at pH 7.5, 1 mM EDTA), and their concentrations were determined using the measured absorbance at 260 nm (NanoDrop, Thermo Scientific) and the extinction coefficient of 5.38 × 106 µM^−1^cm^−1^.

### 2.4. Filter Binding Assay

The ^32^P- labeled 5′-domain RNAs (500 cpm) was incubated at 37 °C for two minutes, with RsuA and RsuAΔN (0–15 µM) in HKM4 buffer (80 mM K-Hepes, pH 7.6, 50 mM KCl, 4 mM MgCl_2_, 6 mM β-mercaptoethanol). These samples were kept on ice until they were loaded onto a dot blot apparatus with nitrocellulose (0.1 µm, GE Healthcare) and nylon (0.45 µm, GE Healthcare) membranes that were soaked in filter binding buffer (20 mM Tris-HCl at pH 7.6, 100 mM CH_3_COOK, 200 mM KCl, 2.5 mM MgCl_2_, and 1 mM DTT). After the passing reaction mixture through membranes under vacuum, membranes were washed thrice with 100 µL of cold filter binding buffer under vacuum. Both membranes were then vacuum dried and exposed to a phosphorimager screen overnight. Radiographs were obtained using a Typhoon FLA 9500 (GE Healthcare). These radiographs were quantified with ImageJ (1.48v) software. The fraction bound was calculated using intensities of the spots in nitrocellulose and nylon membranes for each RsuA concentration (Fraction bound = I_NC_/(I_NC_ + I_Ny_); I_NC_ and I_NY_ are the spot intensity in nitrocellulose and nylon membranes, respectively). The fraction bound versus RsuA concentration plots were fitted to the single binding isotherm (Fraction bound = 1/(1 + K_d_/[RsuA])) to obtain K_d_ for RsuA-RNA complexes.

### 2.5. FRET-Based RsuA Binding Assay

The helix 3 extension of modified 5′-domain RNA (2 nM) was annealed to a 37-nucleotide long DNA oligomer (2 nM), fluorescently labeled with Cy5 at 3′-end as previously described by Abeysirigunawardena et al. [19]. RNAs were then folded in respective Mg^2+^ concentrations (0–20 mM) for 15 min at 37 °C. For RsuA titrations that were carried out in the presence of ribosomal proteins, pre-folded 5′-domain RNAs were co-incubated with a 2.5 molar excess of ribosomal proteins for another 15 min at 37 °C. Fluorescence spectra (560–700 nm) of RNA-protein complexes were recorded upon excitation at 560 nm using a PTI Quantum Master-400 fluorometer (Horiba Scientific). FRET efficiencies at each RsuA concentration were calculated with adjusted fluorescence intensities at 565 nm (I_565_, donor) and 665 nm (I_665_, acceptor) using E = I_665_/(I_665_ + I_565_) equation. Titration curves for triplicates were averaged and fitted to the quadratic equation or a single binding isotherm in Origin software to obtain equilibrium dissociation constants for RNA-protein complexes.

### 2.6. RsuA Activity Assay

The 16S 5′-domain rRNAs (10 pmols) were incubated (at 37 °C for 1 h) with protein RsuA and different combinations of ribosomal proteins S4, S16, S17 and S20 (2.5 molar excess) in HKM4 buffer (80 mM K-Hepes, pH 7.6, 330 mM KCl, 4 mM MgCl_2_, 6 mM β-mercaptoethanol) (Appendix A). A phenol-chloroform extraction was then performed to remove proteins followed by ethanol precipitation. Precipitated RNAs were then dissolved in 10 µL of TE buffer (10 mM Tris-HCl at pH 7.5, 1 mM EDTA). Pseudouridylated RNAs (3 pmols) in 100 µL of BEU buffer (7 M Urea, 4 mM EDTA, 50 mM bicine pH 8.5) were then incubated with 200 mM CMCT (1-cyclohexyl-(2-morpholinoethyl) carbodiimidemetho-p-toluene sulfonate) for 5 min at 37 °C. The CMCT treated samples were then ethanol precipitated twice to remove urea and excess CMCT. CMCT treated RNA was dissolved in 100 µL of sodium carbonate buffer (50 mM Na_2_CO_3_, pH 10.4, 2 mM EDTA) before incubation at 37 °C for 4 h. Alkaline-treated 5′-domain RNAs were precipitated, pelleted, dried, and resuspended in water. SuperScript^TM^ III Reverse Transcriptase (Invitrogen) was used to generate cDNA. The manufacturer suggested protocol was followed for these reverse transcription assays. cDNAs from reverse transcription assays were run on an 8% polyacrylamide denaturing gels for 1 h at 55 W per gel. These gels were dried and then exposed to a phosphorimager screen overnight. Radiographs were then recorded using a Typhoon FLA 9500 (GE Healthcare) scanner. Band intensity of the reverse transcriptase pause at the pseudouridylation site was measured with ImageJ (1.48v) software (NIH). The band intensity at the pseudouridylation site was normalized to the band intensity at position 518 to obtain the normalized activity. Standard error was calculated from three independent replicates.

## 3. Results

### 3.1. Magnesium Ions Influence RsuA Binding

Divalent cations are critical for the folding of many naturally occurring RNAs to their active native structures. RNA-backbone contacts between different regions that are required to generate compact and functional RNA structures are stabilized by magnesium ions [20,21,22,23,24]. RNA-RNA contacts in 30S 5′-domain rRNA to which RsuA binds are formed gradually with the increase in the magnesium concentration. At 10 mM Mg^2+^ concentration, the 30S 5′-domain can form most of its native contacts even in the absence of r-proteins [25]. RsuA may bind to a 5′-domain folding intermediate, in which only certain native contacts are formed. Similarly, r-protein S4 binds to both a folding intermediate of 30S 5′-domain and its native structure with contrasting affinity, while preference for the native and intermediate complexes varied with Mg^2+^ concentration [19]. The stability of RsuA-RNA complexes was determined at varying Mg^2+^ concentrations to investigate the nature of the 5′-domain structure preferred by RsuA. A FRET-based binding assay was performed to determine equilibrium dissociation constants of various RsuA-5′-domain rRNA complexes (Figure 2). In this assay, Cyanine5-labeled RsuA (RsuA-Cy5) was titrated onto fluorescently labeled (Cy3) 5′-domain RNA (21–556 nts), as shown in Figure 2A at various Mg^2+^ concentrations (0-20 mM) [18]. The FRET signal gradually increased and reached a maximum of ~0.03 when RsuA was added, indicating the ability of RsuA to bind to 30S 5′-domain rRNA, even in the absence of r-proteins (Figure 2B). RsuA titration curves were fitted to the single binding isotherm to calculate the dissociation constant (K_d_). The highest affinity of RsuA to 5′-domain rRNA (K_d_ = 0.56 ± 0.04 nM) was observed at 8 mM Mg^2+^ concentration (Figure 2C). At no added magnesium, the K_d_ was found to be ~30-fold higher (19 ± 5 nM) than at 8 mM Mg^2+^, clearly indicating the need for magnesium for stable binding of RsuA. Similarly, K_d_s for RsuA-RNA complexes were found to be approximately 20-fold (11 ± 2 nM) and seven-fold (3.9 ± 0.4 nM) increased at 2 mM and 4 mM Mg^2+^, respectively. At moderate magnesium concentrations (6 mM ≤ Mg^2+^ ≤ 10 mM), RsuA-RNA complexes were found to be highly stabilized at the sub-nanomolar range. However, the stability of RsuA complexes decreased as the Mg^2+^ concentration was increased above 10 mM. Equilibrium dissociation constant increased as the magnesium concentrations were increased from 10 mM–20 mM (7.5 ± 0.4 nM). Similarly, K_d_s at 12 mM and 16 mM were found to be 10 ± 2 nM, 14.7 ± 0.7 nM, respectively. A similar trend of K_d_ with increasing magnesium was observed previously for S4-rRNA complexes, in which the lower affinity for S4 at high magnesium concentration was attributed to charge screening effect of Mg^2+^ ions [26]. However, it is also likely that RsuA defers binding to a 5′-domain when many native contacts are formed.

### 3.2. RsuA Binds Preferably to 16S Helix 18 Pseudoknot Mutants

The 16S helix 18 in which Ψ516 modification is present, forms a pseudoknot structure that is essential to position the universally conserved G530 at the 30S decoding site [7] (Figure 3A,B). Three nucleotides (nts 524–526) at the tip of helix 18 upper hairpin loop (530 loop) base pair with three nucleotides (nts 505–507) in the internal loop at the base of helix 18. Previous single-molecule FRET assays showed that when the tip of helix 18 was deleted, the lack of the ability to form the helix 18 pseudoknot structure caused a 5′-domain assembly intermediate to be stabilized compared to its native complex [18]. The population of the intermediate complex increased at low magnesium concentrations [18,19]. Given the preference of RsuA binding at low magnesium (6–10 mM), we hypothesized that helix 18 mutants, which cannot form the pseudoknot, bind to RsuA with a tighter affinity compared to wild type 16S 5′-domain rRNA. Three pseudoknot mutants—two single nucleotide mutants (G505C, C507G), and a double mutant (G505C:G506)—were constructed using site-directed mutagenesis. A previous study showed that the deletion of helix 18 upper hairpin loop does not influence the 16S secondary structure [18]. Therefore, we assumed that point mutations in the helix 18 sequence would behave similarly. To test the hypothesis that RsuA binds preferably to the extended helix 18 compared to the pseudoknotted, the same FRET-based assay was performed using these three 5′-domain rRNA mutants. Protein RsuA illustrated more than 30-fold higher binding affinity to all three helix 18 mutant 5′-domain RNAs compared to the wild type 5′-domain RNA (Figure 3C). The 5′-domain RNA-RsuA complexes formed with single mutant C507G showed a 390-fold decrease in K_d_ compared to the complex formed with wild type 5′- domain (K_d_ = 3.9 ± 0.4 nM). These results indicate that RsuA preferably binds to assembly intermediates in which helix 18 is extended and not forming a pseudoknotted structure.

### 3.3. Binding of RsuA and S17 Is Thermodynamically Cooperative

During ribosome biogenesis, many ribosomal RNA modification enzymes favorably bind to specific ribosome assembly intermediates compared to fully assembled or naked 16S rRNA [27,28,29]. Such observations indicate that the presence of several selected ribosomal proteins is necessary and essential for modification enzymes to bind to their substrate rRNAs and perform their functions. The ability of ribosomal proteins to preferably stabilize ribosome assembly intermediates suggests the existence of thermodynamic cooperativity between modification enzymes and r-proteins. RsuA pseudouridylates U516 of 5′-domain rRNA only in the presence of r-proteins [5]. For these experiments, the total protein content of the 30S (TP30) was incubated with 5′-domain rRNA (nts 1-678) before incubation with RsuA. Although these experiments illustrated the requirement of r-proteins for RsuA activity, they were unable to determine the exact protein combination that is essential for its binding. Furthermore, binding cooperativities between RsuA and r-proteins or the nature of the intermediate that RsuA preferably binds to cannot be deduced from those experiments. Three primary assembly proteins (S4, S17, and S20) and secondary assembly proteins, (S16 and S12) bind to 16S 5′-domain rRNA. Binding of secondary assembly protein S16 requires the presence of S4 and S20, whereas r-proteins S17, S8, and S5 are required for the stable binding of S12 [30,31] (Figure 4A). The primary assembly protein S4 and secondary assembly protein S12 are the only proteins that form direct contacts with helix 18, in which Ψ516 is located [32,33]. The FRET-based RsuA binding assay was carried out in the presence of different combinations of ribosomal proteins (S4, S16, S17, and S20) at 4 mM Mg^2+^ to understand how r-proteins that bind to the 5′-domain influence the binding and enzymatic activity of RsuA at physiological Mg^2+^ concentrations (Figure 4). Interestingly, RsuA binds to both 5′-domain rRNA and 30S 5′-domain complexed with S4, S16, S17, and S20 (Figure 4B). However, a 3-fold increase in the binding affinity was observed in the presence of all r-proteins (K_d_ = 1.2 ± 0.2 nM) compared to their absence (K_d_ = 3.9 ± 0.4 nM). A lower binding affinity to RsuA was observed when r-protein S4 was bound to the 5′-domain, suggesting a mild binding anti-cooperativity, perhaps due to the overlap of the binding sites of the two proteins. However, the relatively stable RsuA-5′-domain complex (K_d_ = 1.2 ± 0.2 nM) was formed by native 30S 5′-domain that also included protein S4. The affinity of RsuA was highest (K_d_ = 1.0 ± 0.3 nM) in the presence of protein S17 alone. Interestingly, all the 5′-domain complexes that contained S17 illustrated a high affinity to protein RsuA, indicating binding cooperativity between RsuA and S17. Although the affinity of RsuA to 5′-domain rRNA was decreased in the presence of S4 (4.9 ± 0.5 nM), the addition of S17 to RsuA-rRNA-S4 complexes increased its affinity (2.4 ± 0.4 nM). Previous studies on the 5′-domain assembly illustrated that S17 stabilizes non-native assembly intermediate [19,34], in which helix 3 is flipped away from helix 18 compared to native 30S 5′-domain with helix 3 docked at the base of helix 18. Protein RsuA likely binds to the intermediate complex that has the helix 3 positioned away from helix 18.

A previous study by Ofengand and co-workers [5] on RsuA also showed the inability of RsuA to pseudouridylate U516 in 16S rRNA in the absence of r-proteins. Although this study identified that r-proteins are essential for the activity of RsuA, unfortunately, the impact of individual proteins on the pseudouridylation activity of protein RsuA was not identified. In this work, proteins that are required for optimal activity of RsuA enzyme were identified. In our study, RsuA enzymatic activity was measured with a reverse transcriptase-based assay as previously used to detect pseudouridylation (Appendix A) [35,36,37]. Pseudouridylation was carried out by incubating the protein RsuA with various rRNA-protein complexes formed with 16S 5′-domain and r-proteins, S4, S16, S17, and S20. Pseudouridylated RNAs were reacted with CMCT (1-cyclohexyl-(2-morpholinoethyl) carbodiimidemetho-p-toluene sulfonate) that covalently binds to imino nitrogen in guanine (G), uridine (U) and pseudouridine (Ψ) [38]. CMC treated RNAs were then exposed to alkaline conditions (pH = 10.4) to remove all CMC that are bound to guanine and uridines, except pseudouridines (Appendix A) [39]. Finally, the reverse transcriptase pause at U516 is quantified to determine the level of pseudouridylation (Appendix A). As observed by Ofengand and co-workers [35,36], a reverse transcriptase pause was observed at the pseudouridylation site at position 516. However, reverse transcriptase could extend beyond the pseudouridylation site up to the full-length product. Although RsuA binds to the 5′-domain rRNA in the absence of r-proteins (Figure 2B), its pseudouridylation activity required the presence of ribosomal proteins, which is in the agreement of the observation by Ofengand and co-workers. The pseudouridylase activity of RsuA was found to be highest (0.8 ± 0.6) in the presence of protein S17 alone (Figure 4B). However, protein S17, combined with other proteins, also showed high pseudouridylase activity compared to when no proteins were added. For instance, the pseudouridylase activity of RsuA was 0.6 ± 0.3 for both the S4-S17-rRNA complex and the 5′-domain that is complexed with all 5′-domain-binding proteins. Enzymatic activity of RsuA was the highest in the presence of S17, indicating that the flipped intermediate complex may be required for its enzymatic activity.

### 3.4. RsuA Peripheral Domain Increases the Stability of RsuA-rRNA Complexes

Many classes of pseudouridine synthases contain peripheral domains linked to the core domain that possess their catalytic activity [10]. These peripheral domains may not participate in the catalytic activity of pseudouridine synthases; nevertheless, they play a role in substrate recognition [12,13]. In this study, we hypothesize that RsuA peripheral domain is responsible for recognizing the flipped intermediate complex. The N-terminal domain, which includes amino acid residues 1-44, is the peripheral domain in RsuA [8] (Figure 5A). A flexible linker connects the S4-like domain to the core domain of the RsuA protein, allowing its motion with respect to the core domain. X-ray crystal structures of *E. coli* and *Haemophilus influenzae* illustrated that the S4-like domain exists in two different conformations [8,12] (Figure 5A). RsuA S4-like domain could interact with the core domain or can be projected away from the core domain. Filter binding assays were performed using the S4-like domain truncation mutant (RsuAΔN) and the wild-type RsuA protein (Figure 5) to test the hypothesis that the S4-like domain can influence the binding thermodynamics of protein RsuA to 5′-domain rRNA. Signal intensities observed for both nitrocellulose and nylon membranes were quantified, and the fraction bound for each protein concentration (0–15 µM) were calculated (Figure 5B). The fraction bound versus protein concentration plots were fitted to the binding isotherm equation to determine the dissociation constant (Kd). Both RsuA and RsuAΔN bind to the 5′-domain RNA up to 2 µM protein concentration. At high RsuA concentrations, RsuA did not form complexes with rRNA. Unlike the full-length wild type protein, the truncated RsuA protein RsuAΔN remained bound to 16S 5′-domain rRNA at higher protein concentrations (2–15 µM). However, a five-fold decrease in the binding affinity was observed in the absence of the S4-like domain (57 ± 30 nM) compared to its presence (12 ± 4 nM) (Figure 5C), suggesting that the S4-like domain may help protein RsuA for its stable binding. In addition, the S4-like domain may interact with the catalytic domain of RsuA and can prevent its binding to 16S 5′-domain.

## 4. Discussion

Nucleotide modifications in the 16S 5′-domain appear in assembly intermediates prior to other 16S nucleotide modifications. RsuA is likely one of the first modification enzymes that bind to ribosome assembly intermediates during ribosome biogenesis. Previous works by Ofengand and co-workers [5] confirmed that RsuA was likely to be active when it was bound to an assembly intermediate than the native 5′-domain. However, our work illustrates that RsuA can bind to rRNA in the absence of r-proteins, although their activity is known to be dependent on the presence of r-proteins. X-ray crystal structures of the 30S ribosome show the presence of several Mg^2+^ ions near helix 18 and especially close to Ψ516, indicating the importance of Mg^2+^ ions to maintain the structure of the RsuA binding site [19,40,41]. The weaker affinity at lower Mg^2+^ stresses the importance of some structural organization in 16S helix 18 for RsuA binding, especially in the absence of r-proteins. Surprisingly, however, higher concentrations of Mg^2+^ were found to be detrimental to RsuA binding, suggesting that RsuA prefers to bind to a less-compacted structure in which all 16S 5′-domain native contacts are not formed. The pseudoknot mutants’ ability to form stable complexes with RsuA also suggests that RsuA prefers to bind to the extended helix 18 structure compared to the pseudoknotted helix 18. Single-molecule FRET measurements by Kim et al. [18] illustrated that 5′-domain RNAs which were incapable of forming a helix 18 pseudoknot, stabilized a non-native RNA-S4 complexes in which helix 3 is flipped away from the base of helix 18. It is also likely that protein RsuA preferably binds to an assembly intermediate with helix 3 away from helix 18 (Figure 6) [19].

As noted by Ofengand and co-workers [5], RsuA activity requires the presence of r-proteins. Although r-proteins are not necessary and essential for the binding of RsuA, in their presence, RsuA-RNA complexes were stabilized, indicating binding cooperativity between RsuA and r-proteins. Our study shows the existence of binding cooperativity between proteins S17 and RsuA, which suggests that RsuA prefers to bind to an intermediate that is stabilized by S17. Previous studies of 5′-domain assembly [34] have shown that S17 stabilizes an assembly intermediate, in which the base of helix 18 is exposed. At the Mg^2+^ concentrations tested, both S17 and S4 showed contrasting preferences for RsuA binding, which is well complemented by previous hydroxyl radical footprinting assays. Even in the presence of both S4 and S17, the flipped intermediate complex was preferred over the native 5′-domain complex in the agreement of forming stable RsuA-RNA complexes in the presence of both proteins [18,19]. Binding of S4 and RsuA are anti-cooperative, perhaps due to the ability of S4 to stabilize the helix 18 pseudoknot. Not only does the flipped intermediate form stable RsuA complexes, but it is also required for the pseudouridylase activity of RsuA. Even though RsuA can bind to both protein unbound rRNA and S4-rRNA complexes, it cannot pseudouridylate U516 in both rRNAs. Perhaps both S4 and the S4-like domain of RsuA may share the same binding site, thus making unproductive RsuA-rRNA complexes.

Unlike many of the rRNA nucleotide modification enzymes, RsuA may not rely entirely on the sequence-based target recognition. Ribosomal RNA structure at the recognition site may also influence the binding specificity of RsuA. Our data suggest that the N-terminal domain of RsuA may increase its specificity toward 5′-domain RNA, hence play a role in target recognition (Figure 6). This peripheral domain of RsuA is structurally similar to the C-terminal domain of r-protein S4 [8,9]. Interestingly, the S4 C-terminal domain structure is dissimilar in free and RNA-bound protein. Similarly, rRNA may capture a specific conformation of the RsuA S4-like domain out of many possible conformations. In the 30S ribosome, the C-terminal domain of S4 forms contacts with both 16S helix 3 and helix 18. The S4-like domain of RsuA may bind to the junction of helices 3 and 18 followed by formation of contacts between RsuA core domain and helix 18 upper hairpin loop (530 loop). Unlike the mutual induced-fit mechanism previously observed in ribosome assembly [42,43,44], RsuA binding to its target RNA may follow a mutual structure-capture mechanism [44,45]. We suggest that *E. coli* RsuA may exhibit two major conformations with respect to the relative position of the S4-like domain. The S4-like domain may extend away from the core-domain to form an open conformation of RsuA, while RsuA forms a closed conformation when the S4-like domain is bent towards the core-domain. Very high concentrations of RsuA may even promote dimerization. The S4-like domain may create specific rRNA-protein interactions with helix 3 (flipped intermediate) only when RsuA is present in the open conformation. Hence, protein RsuA may recognize the flipped intermediate as its preferred substrate over the native-like 5′-domain complex.

Despite RsuA being a non-essential protein (like some r-proteins) for bacterial survival [6], RsuA may function as an assembly factor that streamlines ribosome biogenesis. During ribosome biogenesis, S17 may bind to its binding site even before the helix 18 and helix 3 is transcribed. Due to the thermodynamic cooperativity that exist, S17 and RsuA may form stable and active complexes and pseudouridylate U516. Bacteria deficient in RsuA do not show growth defects under a range of temperatures and nutrient levels, indicating that the lack of RsuA induces only slight changes to 30S structure; hence, the ribosome functions under conducive conditions [6]. However, the RsuA gene was found to be essential for the survival of MazF toxin-induced cells [46]. Perhaps RsuA is needed for the biogenesis of “specialized ribosomes” that function under stress conditions, and the action of RsuA may eliminate the need for proteins, such as S12, to produce a stable decoding site [47,48].

## 5. Conclusions

In conclusion, our study illustrates binding cooperativity for proteins S17 and RsuA. The peripheral domain of RsuA is needed for its stable binding, and the binding may follow a mutual structure capture mechanism, in which both the protein and rRNA capture a preferred structure.

## Figures and Tables

**Figure 1 biomolecules-10-00841-f001:**
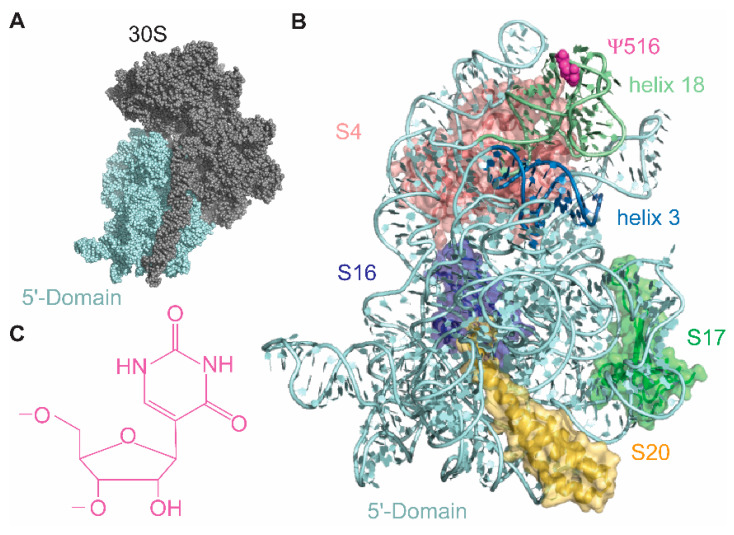
The small subunit (30S) of the *E. coli* ribosome contains a single pseudouridine residue (Ψ516). (**A**) The 30S 5′-domain (cyan) forms the body of the 30S subunit. (**B**) Ribosomal proteins S4 (pink), S17 (green), S16 (blue), and S20 (yellow) bind to the 5′-domain rRNA (cyan); 16S helices 3 and 18 are shown in blue and light green ribbons, respectively. Magenta spherical structure is the Ψ516. (**C**) Chemical structure of pseudouridine.

**Figure 2 biomolecules-10-00841-f002:**
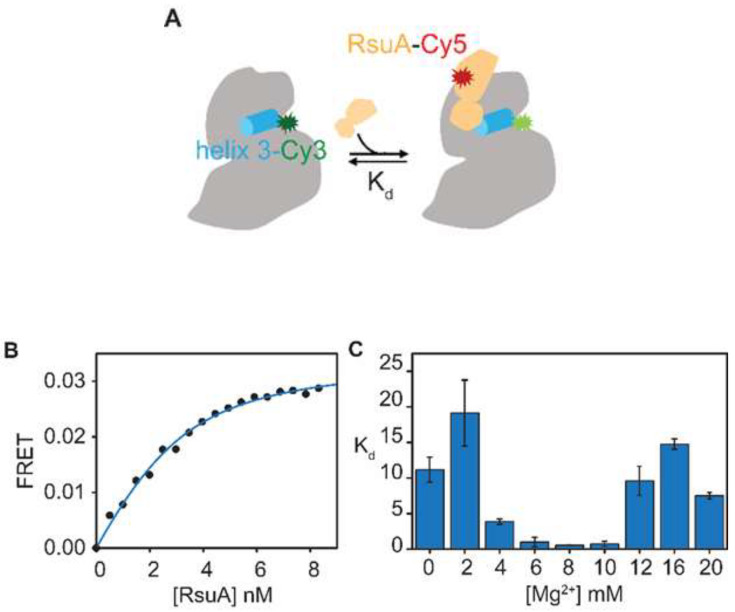
Protein ribosomal small subunit pseudouridine synthase A (RsuA) binds to the 30S 5′-domain. (**A**) Florescence labeling scheme used for FRET-based RsuA binding assays is shown. Helix 3 (cyan) of the 30S 5′-domain (gray) is annealed with a DNA primer conjugated to Cy3 dye (green), whereas protein RsuA (wheat) is labeled with Cy5 dye (red) at Cys127. (**B**) A representative RsuA titration curve at 4 mM [Mg^2+^]. (**C**) Binding affinities of protein RsuA to the 5′-domain of 16S rRNA in various Mg^2+^ concentrations (0–20 mM) are shown. RsuA titrations were performed in HKM_X_ buffer (80 mM K-Hepes pH 7.6, 330 mM KCl, and 0–20 mM MgCl_2_). Error bars represent errors obtained by the least-square curve fitting.

**Figure 3 biomolecules-10-00841-f003:**
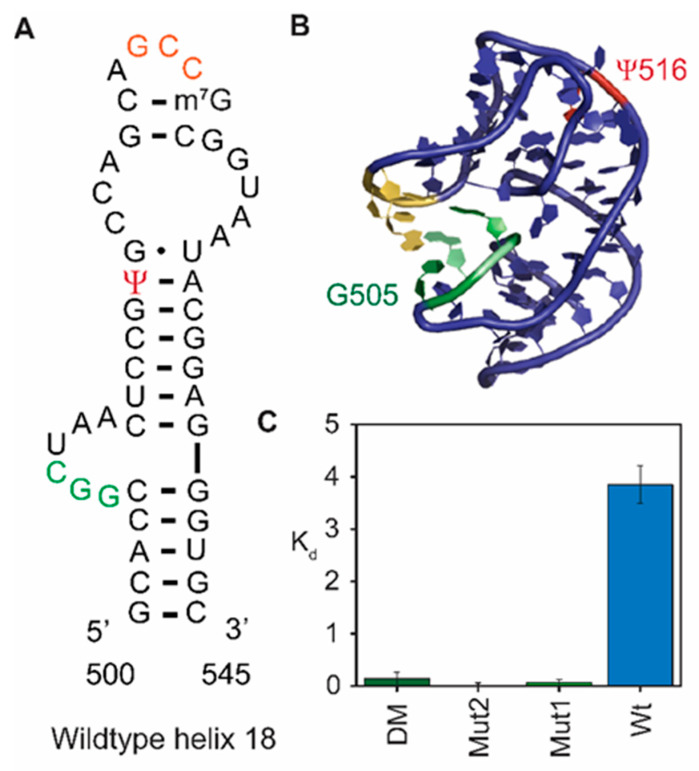
Protein RsuA preferably binds to helix 18 pseudoknot mutants compared to wild type 5′-domain. *E. coli* 16S rRNA helix 18 (**A**) secondary and (**B**) Three-dimensional structures are shown. Pseudouridine modification at position 516 is shown in red. Nucleotides 505–507 (green) form base pairs with nucleotides 524–526 (yellow) to form the helix 18 pseudoknot. (**C**) Binding affinities of protein RsuA to wild type 5′-domain and helix 18 mutants at 4 mM Mg^2+^ are shown.

**Figure 4 biomolecules-10-00841-f004:**
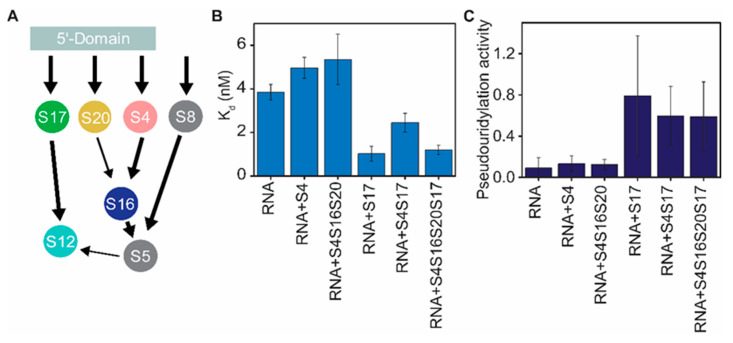
Binding affinity and enzymatic activity of protein RsuA vary with different combinations of ribosomal proteins. (**A**) The Nomura map of 30S 5′-domain assembly. S12 binding also requires the presence of S8 (gray) and S5 (gray) that bind to the 30S central domain. (**B**) Binding affinities and (**C**) Pseudouridylase activities of protein RsuA (4 mM MgCl_2_) in the presence of various combinations of proteins S4, S17, S20, and S16. Error bars represent propagated errors for biological triplicates.

**Figure 5 biomolecules-10-00841-f005:**
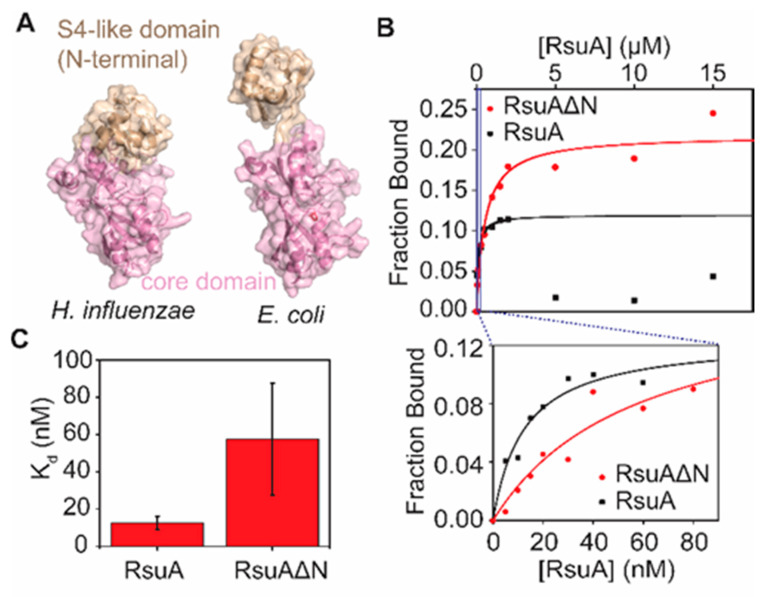
The S4-like domain of protein RsuA helps protein RsuA to bind with rRNA. (**A**) X-ray crystal structures of *Haemophilus influenzae* and *Escherichia coli* RsuA. RsuA S4-like domain and core domains are shown in wheat and pink, respectively. (**B**) Binding isotherms obtained for RsuAΔN (red circles) and wild type RsuA (black squares) from filter binding assays are shown. For wild type RsuA, the bound fraction decreased at concentrations higher than 2 μM, perhaps due to dimerization of wild type RsuA. Black (wild type RsuA) and red (RsuAΔN) lines represent least-square fitting for 0–2 μM and 0–15 μM range, respectively. (**C**) Binding affinities of wild type RsuA and S4-like domain truncated RsuAΔN to the 30S 5′-domain. The average of three replicates is shown. The error bars represent propagated errors.

**Figure 6 biomolecules-10-00841-f006:**
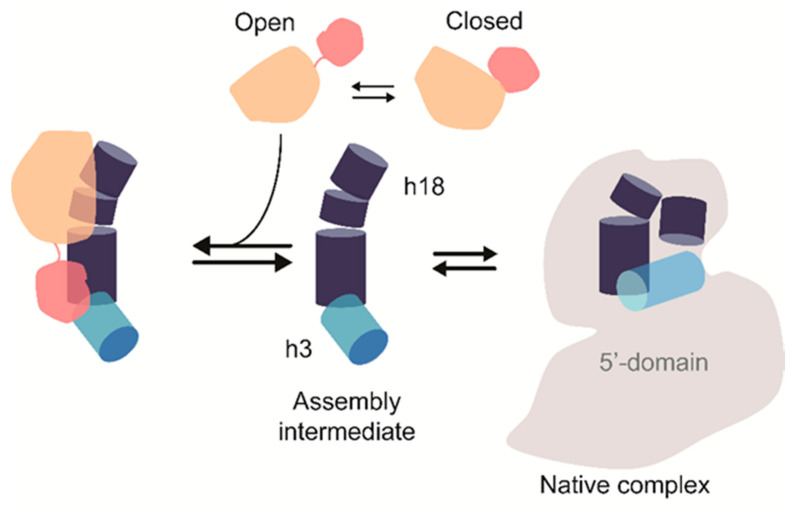
The current working model for RsuA binding is shown.

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
