# Peer review of "Pseudouridine Synthase RsuA Captures an Assembly Intermediate That Is Stabilized by Ribosomal Protein S17"

_biomolecules, 2020, doi:10.3390/biom10060841_

Round 1

Reviewer 1 Report

In the manuscript by Jayalath et al., the authors perform biochemical assays with purified recombinant RsuA to evaluate its binding to an RNA substrate representing its known target in the 30S ribosomal subunit as well as psedouridinylation activity under different Mg2+ concentrations and in the presence of various ribosomal proteins. The results are suggestive of a better binding to an rRNA assembly intermediate rather than native or fully matured rRNA, further enhanced by ribosomal proteins in the vicinity. Due to the limited nature of these experiments, the results do not represent the definitive, final say on the subject, but are generally supportive of the authors' argument. To improve readability, a fairly intensive English language editing of the manuscript is recommended prior to publication.

20-22: "Our current working model suggests that both protein RsuA and its substrate RNAs exist in multiple conformations and RsuA may follow mutual structure capture mechanism for its substrate recognition." I was lost as to what this might mean. Having a cryptic phrase like that in the abstract is not helpful. Please re-write. Perhaps using here the concluding phrases in the paper (lines 402-404) would make the paper's conclusions easier to understand when reading the abstract.

47: Jumping to Fig 5 right after Fig 1 is confusing and probably unnecessary, Fig. 5 does not add much to what is discussed the text.

89-97, 109, 129, 157: What is the pH of the Hepes buffer and what base was used to titrate Hepes?

166: The authors just generate cDNA rather than construct a library.

169: I assume the authors meant phosphorimaging screen, not an intensifying screen.

262: "fully assembled 30S 5′-domain complexed with S4, S16, S17, and S20". The authors do not show that the domain is fully assembled, only perform the reaction in the presence of different protein combinations. Please rephrase accordingly.

Fig. 5b: the axis label at the top can be misinterpreted as the ratio of the two proteins.

Fig. 5b: Lack of RsuA binding to RNA at higher concentrations is somewhat worrisome. The authors mention dimerization later in the article, could this be the issue? Whatever the reason is, the curve should not be fitted as shown (i.e., going to a saturation). At a minimum, please show the curve that fits the data and (in a dashed line) the approximation used for the Kd calculations. The low concentration range should be shown for both wild type RsuA and its truncation mutant.

Author Response

We thank the editorial board and all reviewers for their valuable time and effort to provide us thorough revisions and constructive feedback during these days of uncertainty. We believe the quality of our manuscript was greatly improved after these revisions. We incorporated all but one changes suggested by reviewers. Again, we greatly appreciate reviewers for their insightful comments and for giving us the opportunity to submit a revised draft of the manuscript. 

Referee 1

Recommendation: Intensive English language editing is recommended

Thank you for pointing out. We used Grammarly to correct any grammatical errors and typos. We significantly modified the text to improve clarity. In addition, the text was reviewed/proofread by two native English speakers after the corrections were made.    

In the manuscript by Jayalath et al., the authors perform biochemical assays with purified recombinant RsuA to evaluate its binding to an RNA substrate representing its known target in the 30S ribosomal subunit as well as psedouridylation activity under different Mg2+ concentrations and in the presence of various ribosomal proteins. The results are suggestive of a better binding to an rRNA assembly intermediate rather than native or fully matured rRNA, further enhanced by ribosomal proteins in the vicinity. Due to the limited nature of these experiments, the results do not represent the definitive, final say on the subject, but are generally supportive of the authors' argument. To improve readability, a fairly intensive English language editing of the manuscript is recommended prior to publication.

To improve readability, a fairly intensive English language editing of the manuscript is recommended prior to publication.

Thank you very much for your suggestion. We corrected all grammatical mistakes throughout the manuscript. In addition, several changes were made to improve the language and make the manuscript more legible for readers.

20-22: "Our current working model suggests that both protein RsuA and its substrate RNAs exist in multiple conformations and RsuA may follow mutual structure capture mechanism for its substrate recognition." I was lost as to what this might mean. Having a cryptic phrase like that in the abstract is not helpful. Please re-write. Perhaps using here the concluding phrases in the paper (lines 402-404) would make the paper's conclusions easier to understand when reading the abstract.

21-23: Thank you very much for pointing out! We fixed the abstract. We hope now it reads better than the first submission.

47: Jumping to Fig 5 right after Fig 1 is confusing and probably unnecessary, Fig. 5 does not add much to what is discussed in the text.

49 (45): We agree with the comment and removed the fig 05a from line 47. Thank you for the suggestion!

89-97, 109, 129, 157: What is the pH of the Hepes buffer and what base was used to titrate Hepes?

94, 97, 100, 102, 116, 137, 166 and 226 (89, 92, 94, 97, 110, 129, 157 and 213): Sorry for the omission. Hepes buffers had a final pH of 7.6. Please note the changes in the text.

166: The authors just generate cDNA rather than construct a library.

175 (166): Thank you for pointing out the mistake—changed cDNA library to cDNA

169: I assume the authors meant phosphorimaging screen, not an intensifying screen.

179 (169): Yes, we meant the phosphorimaging screen, and it was fixed in the manuscript. Thanks!

262: "fully assembled 30S 5′-domain complexed with S4, S16, S17, and S20". The authors do not show that the domain is fully assembled, only perform the reaction in the presence of different protein combinations. Please rephrase accordingly.

263 (277): We apologize for the lack of clarity. It has been reported that the 30S 5′ domain RNP assembles completely under physiological conditions. (Ramaswamy, P.; Woodson, S.A. S16 throws a conformational switch during assembly of the 30S 5' domain, Nat. Struct. Mol. Biol, 2009, 16(4), 438-445. Kds for all 5'-domain-binding proteins are below the range of the concentrations of the proteins used. Therefore, under the reaction conditions used, we assumed that 5'-domain complexes were fully assembled in the presence of S4, S16, S17, and S20. However, the sentence was rearranged. Thank you!

Fig. 5b: the axis label at the top can be misinterpreted as the ratio of the two proteins.

Fig. 5b: Thank you so much for pointing that out. Fixed it

Fig. 5b: Lack of RsuA binding to RNA at higher concentrations is somewhat worrisome. The authors mention dimerization later in the article, could this be the issue? Whatever the reason is, the curve should not be fitted as shown (i.e., going to a saturation). At a minimum, please show the curve that fits the data and (in a dashed line) the approximation used for the Kd calculations. The low concentration range should be shown for both wild type RsuA and its truncation mutant.

Fig 5b: Thank you for pointing out, and we are sorry for the confusion. We fixed the axes label to avoid any confusion. We only considered the data points obtained up to 2 μM for our fitting of the wildtype curve. We tried many methods to draw a dashed line for part of the fitting curve without any success. Therefore, the figure legend was modified to include data ranges used for the fitting. Now the short-range titrations are shown for both wildtype and mutant RsuA as requested. 

Reviewer 2 Report

Jayalath et al. report on ribosomal RNA pseudouridylating enzyme RsuA and the role various components of the ribosome (rproteins and rRNA helices) for modification to occur. The authors have used a variety of biophysical and biochemical techniques to probe the interactions and folding of the rRNA helices and modification status. The results are clearly reported and the discussion is well done. The authors provide some new insight into the role of cooperativity of protein binding and regulation of rRNA conformation states (or trapping of conformational states) in order to achieve efficient modification of U516 (i.e., pseudouridylation).

Minor comments:

The gel in Figure S3 needs some labels on the nucleotide positions (it is unfortunate that there are no sequencing lanes, but under the circumstances with lab closures, etc., I think the data shown are sufficient).

The sentence on p. 10, line 366 is not clear. 

The manuscript needs editing due to grammatical errors. Some suggestions are included in the marked document.

Author Response

We thank the editorial board and all reviewers for their valuable time and effort to provide us thorough revisions and constructive feedback during these days of uncertainty. We believe the quality of our manuscript was greatly improved after these revisions. We incorporated all but one changes suggested by reviewers. Again, we greatly appreciate reviewers for their insightful comments and for giving us the opportunity to submit a revised draft of the manuscript. 

Referee 2

Recommendation: Minor comments

Jayalath et al. report on ribosomal RNA pseudouridylating enzyme RsuA and the role various components of the ribosome (rproteins and rRNA helices) for modification to occur. The authors have used a variety of biophysical and biochemical techniques to probe the interactions and folding of the rRNA helices and modification status. The results are clearly reported and the discussion is well done. The authors provide some new insight into the role of cooperativity of protein binding and regulation of rRNA conformation states (or trapping of conformational states) in order to achieve efficient modification of U516 (i.e., pseudouridylation).

The gel in Figure S3 needs some labels on the nucleotide positions (it is unfortunate that there are no sequencing lanes, but under the circumstances with lab closures, etc., I think the data shown are sufficient).

Figure S3: Thank you very much for your suggestion and kind understanding of the impact of the Covid19 pandemic on our current research work (labs are closed with no set date for reopening). Labels of nucleotide positions were added to Figure S3. The exact position of Ψ516 was confirmed by G/U lane (CMCT treatment, without alkaline treatment).

The sentence on p. 10, line 366 is not clear.

370-373 (392-395): The sentence is fixed. Thank You!

The manuscript needs editing due to grammatical errors. Some suggestions are included in the marked document.

Thank you very much for your helpful suggestions. We used Grammarly to fix grammar issues.

Reviewer 3 Report

In this manuscript, Jayalath et al. found that Mg2+ concentration can influence RsuA binding and RsuA binds preferably to 16S helix 18 assembly intermediates without a pseudoknotted structure. The authors further showed that the cooperativity of binding and modification activity between r-protein S17 and RsuA. At last, the authors showed that the RsuA S4-like domain contributes to the stability of RsuA-rRNA complexes. These results are interesting and informative on the RsuA functions in ribosome biogenesis. However, a few questions need to be addressed:

  1. Figure 2. First, since the authors showed that RsuA-RNA complexes were highly stabilized at the moderate magnesium concentrations (6-10 mM) in Figure 2c, why the authors used a lower concentration of Mg2+ at 4 mM in the RsuA titration assay? Second, it is better to show the titration results with RsuA at different [Mg2+], which will be more informative. Third, does a chelating agent, such as EDTA block the effect of Mg2+ ions during RsuA binding? The competitive inhibition assay should be performed to further validate the specificity of effect of Mg2+
  2. Again, the authors performed the binding affinities assay using 4 mM other than 6-10 mM Mg2+. The authors should briefly make an explanation.
  3. Figure 3. The authors constructed three pseudoknot mutants to verify that RsuA preferably binds to assembly intermediates. How did the authors confirm that these three mutations only destruct the pseudoknot formation? My concern is that the difference of binding affinity can also result from the global change of secondary or 3-D structure of helix 18, not just from the failure of pseudoknot formation.
  4. Figure S3c. Although the authors labeled the pause site at position 516 with red box, multiple bands were detected from the gel image. I suggested the authors to run the RT-products alone with the RNA for the A, G, C, U sequencing gel, which will provide more convincing evidence.
  5. Figure 5c. The concentration for the Y axis is not consistent with the main text (line 327): for example, ~10 nM in the Figure versus 0.1 nM for the wildtype RsuA in the text of manuscript.

Formatting mistakes

For example:

  1. line 371: “RsuA may not be rely entirely on…” should be “RsuA may not rely entirely on…”
  2. line 211: 4mm should be 4mM
  3. Acknowledgements: line 416, “In” should be deleted.

Conflicts of Interest: line 418, “Declare” at the beginning of sentence should be deleted.

  1. References: All the references should be carefully edited according to the format of the journal. For example:

Ref. 4:  line 427, “Trends Biochem Sci” should be “Trends Biochem. Sci.”;

Ref. 12:  line 448, “Acta. Cryst” should be “Acta. Cryst.;

I suggest the authors to use EndNote or the other software tools to create a reference list.

Author Response

We thank the editorial board and all reviewers for their valuable time and effort to provide us thorough revisions and constructive feedback during these days of uncertainty. We believe the quality of our manuscript was greatly improved after these revisions. We incorporated all but one changes suggested by reviewers. Again, we greatly appreciate reviewers for their insightful comments and for giving us the opportunity to submit a revised draft of the manuscript. 

Recommendation: A few questions are required to be addressed 

In this manuscript, Jayalath et al. found that Mg2+ concentration can influence RsuA binding and RsuA binds preferably to 16S helix 18 assembly intermediates without a pseudoknotted structure. The authors further showed that the cooperativity of binding and modification activity between r-protein S17 and RsuA. At last, the authors showed that the RsuA S4-like domain contributes to the stability of RsuA-rRNA complexes. These results are interesting and informative on the RsuA functions in ribosome biogenesis. However, a few questions need to be addressed:

Figure 2. First, since the authors showed that RsuA-RNA complexes were highly stabilized at the moderate magnesium concentrations (6-10 mM) in Figure 2c, why the authors used a lower concentration of Mg2+ at 4 mM in the RsuA titration assay?

Figure 2: We are sorry for not providing a proper explanation for considering 4 mM [Mg2+] to conduct RsuA titrations in the presence of ribosomal proteins. We selected 4 mM [Mg2+] over 6-10 mM range, since 4 mM is more closer to the physiological magnesium concentration in E.coli cells. In addition, many previous studies provide details on 16S rRNA assembly at physiological magnesium concentration. Unfortunately, the specific assembly details available for 6-10 mM is less compared to 4 mM. Therefore, we considered that obtaining the RsuA binding data at 4 mM [Mg2+] is more appropriate as it allows us to understand and interpret them better by comparing it with the existing data.

Second, it is better to show the titration results with RsuA at different [Mg2+], which will be more informative.

Figure 2: We are sorry for the lack of clarity in presentation. RsuA titrations were performed at different [Mg2+] and are shown in figure 2C.

Third, does a chelating agent, such as EDTA block the effect of Mg2+ ions during RsuA binding? The competitive inhibition assay should be performed to further validate the specificity of the effect of Mg2+

Figure 2: Thank you for your suggestion. The addition of EDTA can decrease the effective concentration of Mg2+ in solution. Since we are considering equilibrium measurements, I believe that there will only be the effect of Mg2+ concentration change. Since we have done titrations at various Mg2+ concentrations (Figure 2c), not much can be gained from adding EDTA. If the concern is the accuracy of [Mg2+] due to the leaching out of Mg2+ from transcription reaction, we want to point out that we perform multiple washes during ethanol precipitation and pass-through size exclusion columns after RNA purification to remove excess salts.  

Figure 3. The authors constructed three pseudoknot mutants to verify that RsuA preferably binds to assembly intermediates. How did the authors confirm that these three mutations only destruct the pseudoknot formation? My concern is that the difference of binding affinity can also result from the global change of secondary or 3-D structure of helix 18, not just from the failure of pseudoknot formation.

Figure 3: Sorry for the lack of clarity. As previously reported by Kim et al. (ref 18), deletion mutations of helix 18 upper hairpin do not have a global effect on the 16S secondary structure as confirmed by a SHAPE assay. Therefore, we assumed that point mutations in the helix 18 sequence would behave similarly. As noted in the text, these mutants cannot form pseudoknot; hence favor a non-native intermediate shown in figure 6.

Figure S3c. Although the authors labeled the pause site at position 516 with red box, multiple bands were detected from the gel image. I suggested the authors to run the RT-products alone with the RNA for the A, G, C, U sequencing gel, which will provide more convincing evidence.

Figure S3c: Thank you for your suggestion, and we do agree with you that the lack of sequencing lanes makes our data less convincing. We confirmed the position of pseudouridylation site, by running RT products along with sequencing lanes and CMC modified RNAs that create RT stops at each U and G. We identify the Ψ516 using the G/U sequencing lane obtained by CMCT treatment. As mentioned previously, bands below the red box could arise from high salts, or urea leached from pseudouridylation and CMCT reaction, respectively. However, for your consideration, we have included an image we have obtained with sequencing lanes. We will be happy to use the gel image from Figure S3C with this image.

Figure 5c. The concentration for the Y axis is not consistent with the main text (line 327): for example, ~10 nM in the Figure versus 0.1 nM for the wildtype RsuA in the text of manuscript.

Figure 5: Thank you so much for pointing this. The numbers in the figures were accurate. We modified the text with accurate numbers (lines 328-329 in revised manuscript or 346-347 with “track changes” function). 

Formatting mistakes

I suggest the authors to use EndNote or the other software tools to create a reference list.

397, 224, 443 and 445 (375, 211, 420 and 422) It is greatly appreciated that you have pointed out formatting errors one by one. They were all corrected in the text. The reference list was edited using reference management software (Mendeley). Thank you so much for your helpful comments and suggestions. 

Round 2

Reviewer 3 Report

The authors addressed almost all my concerns and questions. I suggest that the authors add their explanations for my questions in the text, so the readers will better understand their experiments. 

I did not see the updated image with sequencing lanes. Please make sure to upload the correct version.

After the minor revisions, I am happy to suggest accepting the manuscript for publication.

Author Response

Thank you for reviewing the manuscript again! We apologize for the omissions!!

Referee 3

Figure 2. First, since the authors showed that RsuA-RNA complexes were highly stabilized at the moderate magnesium concentrations (6-10 mM) in Figure 2c, why the authors used a lower concentration of Mg2+ at 4 mM in the RsuA titration assay?

We are sorry for not providing a proper explanation for considering 4 mM [Mg2+] to conduct RsuA titrations in the presence of ribosomal proteins. We selected 4 mM [Mg2+] over 6-10 mM range, since 4 mM is more closer to the physiological magnesium concentration in E.coli cells. In addition, many previous studies provide details on 16S rRNA assembly at physiological magnesium concentration. Unfortunately, the specific assembly details available for 6-10 mM is less compared to 4 mM. Therefore, we considered that obtaining the RsuA binding data at 4 mM [Mg2+] is more appropriate as it allows us to understand and interpret them better by comparing it with the existing data.

Line 560-561: Changes were made to the text to rationalize the use of 4 mM Mg2+. Thank you!

Figure 3. The authors constructed three pseudoknot mutants to verify that RsuA preferably binds to assembly intermediates. How did the authors confirm that these three mutations only destruct the pseudoknot formation? My concern is that the difference of binding affinity can also result from the global change of secondary or 3-D structure of helix 18, not just from the failure of pseudoknot formation.

Sorry for the lack of clarity. As previously reported by Kim et al. (ref 18), deletion mutations of helix 18 upper hairpin do not have a global effect on the 16S secondary structure as confirmed by a SHAPE assay. Therefore, we assumed that point mutations in the helix 18 sequence would behave similarly. As noted in the text, these mutants cannot form pseudoknot; hence favor a non-native intermediate shown in figure 6.

Line 507-509: We apologize for the omission! Reasons were included in the text.        

Figure S3c. Although the authors labeled the pause site at position 516 with red box, multiple bands were detected from the gel image. I suggested the authors to run the RT-products alone with the RNA for the A, G, C, U sequencing gel, which will provide more convincing evidence.

Thank you for your suggestion, and we do agree with you that the lack of sequencing lanes makes our data less convincing. We confirmed the position of pseudouridylation site, by running RT products along with sequencing lanes and CMC modified RNAs that create RT stops at each U and G. We identify the Ψ516 using the G/U sequencing lane obtained by CMCT treatment. As mentioned previously, bands below the red box could arise from high salts, or urea leached from pseudouridylation and CMCT reaction, respectively. However, for your consideration, we have included an image we have obtained with sequencing lanes. We will be happy to use the gel image from Figure S3C with this image.

Figure S3: Explanation was added to the figure legend. Nucleotide positions are now indicated on the gel to make it easy to interpret. A new panel is added to show the comparison of sequencing lanes and CMCT sequencing lane.

Thank you again